# A fractal kinetics SI model can explain the dynamics of COVID-19 epidemics

**Kosmas Kosmidis**[1,2]*, **Panos Macheras**[2]

**1** Division of Theoretical Physics, Department of Physics, Aristotle University of Thessaloniki, Thessaloniki, Greece, **2** PharmaInformatics Unit, Research Center ATHENA, Athens, Greece

* kosmask@auth.gr

## Abstract

The COVID-19 pandemic has already had a shocking impact on the lives of everybody on the planet. Here, we present a modification of the classical SI model, the Fractal Kinetics SI model which is in excellent agreement with the disease outbreak data available from the World Health Organization. The fractal kinetic approach that we propose here originates from chemical kinetics and has successfully been used in the past to describe reaction dynamics when imperfect mixing and segregation of the reactants is important and affects the dynamics of the reaction. The model introduces a novel epidemiological parameter, the "fractal" exponent $h$ which is introduced in order to account for the self-organization of the societies against the pandemic through social distancing, lockdowns and flight restrictions.

**Data Availability Statement:** The data underlying the results presented in the study are available from https://www.tableau.com/covid-19-coronavirus-data-resources and also included as supplementary material.

## Introduction

As of March 30, 2020, coronavirus disease 2019 (COVID-19) has been confirmed in 782,213 people worldwide, leading to 37,579 deaths. Its mortality is apparently higher compared with a mortality rate of less than 1% from influenza [1]. There is an urgent need to model the growth of COVID-19 worldwide. The classical epidemiological approach for the study of growth relies on the reproduction number and infection time, which leads to an exponential growth. However, the data accumulated so far indicate deviation of growth from this pattern. Various approaches to model COVID-19 epidemics have been published in the literature recently based on various hypotheses [2, 3] Our model follows the principles of fractal kinetics [4] which is suitable for chemical reactions under "topological constraints" e.g. insufficient mixing of the reactant species. In fact, there is a full analogy between the governments' measures to ensure social distancing for the control of epidemics and the reactions taking place in low dimensions or insufficient stirring [4–6]

## The model

The classical SI Model (or SIS model as is often referred in the literature [7, 8]) of epidemic spreading is the simplest approach in the mathematical modeling of epidemics [9, 10]. A population comprises two classes, susceptible and infected. The fraction of susceptible individuals is denoted by $S$ and that of the infected is denoted by $I$. The total number of individuals is

**Funding:** The authors acknowledge financial support from "Athena" Research Center for the open access publication of this work.

**Competing interests:** The authors have declared that no competing interests exist.

assumed to be constant and consequentially the sum of the fractions is equal to one, i.e. $S + I = 1$. Thus, the SI model is essentially reduced to a single non linear ordinary differential equation

$$\frac{dI}{dt} = aI(1 - I) - bI \tag{1}$$

where $a$ is proportional to the probability of an infected individual to infect a health one and $b$ is the recovery rate of infected individuals. In the standard SI model it is common to write the $a$ constant per person, i.e. $a/N$ where $N$ is the total size of the population but for our present investigation the form of Eq 1 is adequate.

There are several drawbacks in this model. Most importantly the parameter $a$ is considered a constant, while as recent experience has shown in global epidemics societies tend to organize, governments take measures to ensure social distancing etc. The parameter $a$ is proportional to the number of contacts sufficient to transmit infection per unit time. It is natural to assume that this number is not a constant but a decreasing function of time as a result of two factors. The first is that an infected individual has a finite circle of social contacts and is not surrounded from an infinite pool of healthy individuals. Thus, although initially an infected individual may create a large number of secondary infections as time progresses this number will be lower simply because of the reduction of the number of healthy individuals in his proximity. The second, as the COVID-19 case has shown, is that as soon as a serious threat on public health is identified, measures are taken to promote social distancing, the use of gloves, masks, antiseptics etc. It would be helpful to take the above into account in our mathematical models.

Thankfully, such a challenge is not completely new as a similar situation is often encountered in chemical reaction kinetics. Again, the classical reaction kinetics models assume homogeneity of the reaction medium and of the spatial distribution of the reactants and model chemical reactions assuming reaction rate constants that are independent of time. This approach has, however, shown to be insufficient in heterogeneous media [4] or even in cases where the reactants are not "well-stirred" and, thus, a depletion zone [11, 12] and the reaction sites which leads to an overall slowdown of the total reaction.

The fractal kinetic approach that we propose here originates from chemical kinetics and has successfully been used in the past to describe reaction dynamics when imperfect mixing and segregation of the reactants is important and affects the dynamics of the reaction [4, 11, 13]. A fractal kinetics approach consists in assuming that the reaction constants are in fact functions of time. In particular the assumed form is obtained if $a \rightarrow a/t^h$. The power law form of the function and the "fractal" exponent $h$ is the reason of the term "Fractal Kinetics" proposed by Kopelman in his famous article [4]. Obviously, the limiting case $h = 0$ reduces to classical constant rate kinetics, while values of $h > 0$ arise due to system heterogeneity or/and the creation of a depletion zone around the reactants due to imperfect stirring.

Thus, the fractal kinetics SI model for epidemic spreading is described by the following equation

$$\frac{dI}{dt} = \frac{a}{t^h} I(1 - I) - bI \tag{2}$$

Again, the limiting case $h = 0$ corresponds to the classical SI model. Values of $h > 0$ are expected as a result of measures taken to promote social distancing and public awareness. High values of the $h$ exponent signify considerably different dynamics than those predicted from the classical model and imply a greater influence of public awareness on the disease spreading.

Here, we will estimate the $h$ exponent using publicly available data obtained from [14]. For convenience the data are included in the supplementary materials section. Although it is, in

principle, possible to fit the above equation to the available data for the spreading of the COVID-19 epidemic and calculate the three parameters $a$, $b$, $h$, in practice this approach is rather difficult since Eq 2 can only be solved numerically and numerical instabilities when $t$ is close to zero cause problems to the fitting process. A better approach is to consider the cumulative fraction of infected individuals $I_T$ as a function of time. The differential equation describing this quantity according to the Fractal Kinetic SI model would be similar to Eq 2 with the omission of the recovery term, thus

$$\frac{dI_T}{dt} = \frac{a}{t^h}I_T(1 - I_T) \tag{3}$$

This equation cal be solved analytically and one obtains the rather elegant result

$$I_T = \frac{1}{1 + c\exp\left(\frac{at^{1-h}}{h-1}\right)} \tag{4}$$

This equation can be fitted to the confirmed cases for each country as a function of time and as we present below the fitting is excellent in all cases. Here, we focus on the study of the exponent $h$ which is the novel significant parameter introduced in the scope of Fractal kinetics. High values of the exponent $h$ signify high level of self-organization of the system, while the value of $h = 0$ reduces the model to the classical SI model where perfect mixing of the population is assumed. The parameter $c$ is of considerable importance as it determines the asymptotic limit of $I_T$ i.e. the total fraction of individuals that will be infected. Taking the limit of Eq 4 when $t \to \infty$ we find

$$\lim_{t\to\infty} I_T(t) = \begin{cases} 1 & h \le 1 \\ \frac{1}{1+c} & h > 1 \end{cases} \tag{5}$$

It is well known, however, that most countries test only for serious symptomatic cases and thus the number of confirmed cases seriously underestimates the actual fraction of infected people. The parameters $a$ and $c$ of the model are rather sensitive in such systematic underestimation while the parameter $h$ is rather robust. In order to demonstrate that we have estimated the $a$, $c$, $h$ parameters for the actual data for Greece and then we have multiplied the data by $10^3$ and recalculated them. We found that $c$ changed by a factor of approximately $10^3$, the relative of $a$ was almost 40% while the exponent $h$ changed only by roughly 7% i.e from 1.37 to 1.27.

## Results

Fig 1 shows results for the confirmed cases of nine countries. In each case the confirmed cases are normalized by dividing them by the country population. For each country day zero is considered to be the day of the first confirmed case. Points are the COVID-19 data and the solid line is the model's best fit of Eq 4 to the data [14].

Fig 2 shows countries with more that 1000 confirmed cases ranked in a decreasing order of the $h$ exponent as calculated using outbreak data up to the 29 March 2020. The "fractal" exponent $h$ quantifies the self-organization of the societies against the pandemic through social distancing, lockdowns and flight restrictions. Thus, high values of the $h$ exponent is an indication for the successful application of preventive measures for a country.

Australia and South Korea are ranked on top with practically the same $h$ exponent. For South Korea this is in agreement with the recognition this country has received from the

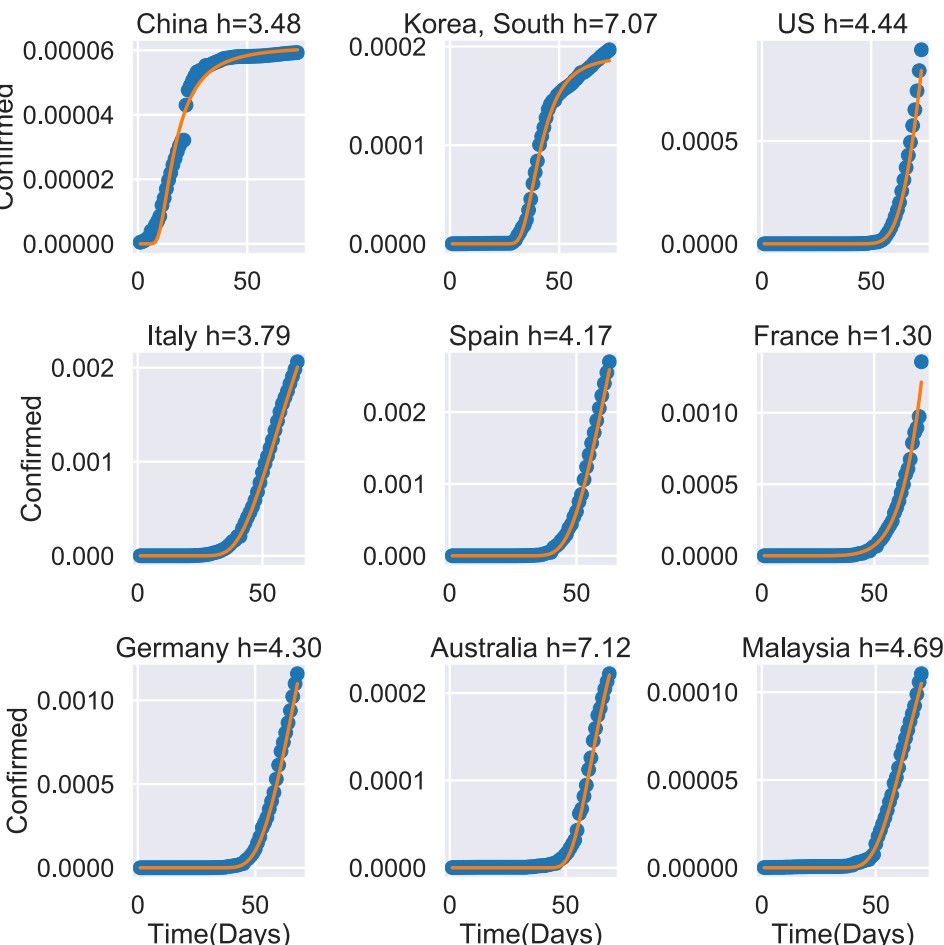

**Fig 1. Confirmed cases of COVID-19, $I_T$ as a function of time for 9 countries.** Points are the actual data [14]. The solid line is the best fit of of Eq 4. Data up to 4-4-2020.

international press on the way that has reacted to the COVID-19 epidemic outbreak [15]. For Australia the high *h* value is probably a result of the country's preventive measures.

We expect that the calculated values of the *h* exponent will change as new data become available and as measures of increasing "social distancing" become effective. Due to the long time of the virus incubation period it is believed that "social distancing" measures taken today will have an observable effect in the number of confirmed cases in 2 weeks. Interestingly, using data up to 29-3-2020 the estimated *h* exponent for Italy is equal to 2.76 while using the latest data (up to 4-4-2020) the Italian *h* exponent increases to 3.79 after the dramatic increase in outbreak cases in Italy and the extreme prevention measures taken by the Italian government [16].

Here we proposed a simple model based on fractal kinetics that is in excellent agreement with the published COVID-19 outbreak data. Of course more detailed modeling could reveal more aspects of the outbreak and lead to a better understanding. We feel, however, that the exponent *h* of the fractal kinetic SI model is a novel, and easily determined parameter, from an epidemiological point of view and, thus, can provide new insight to the disease dynamics. Moreover, due to the excellent fit of Eq 4 to the data for all countries, we believe that even more detailed models would ultimately lead to an expression for the fraction of infected individuals $I_T$ that is similar (if not identical) to Eq 4.

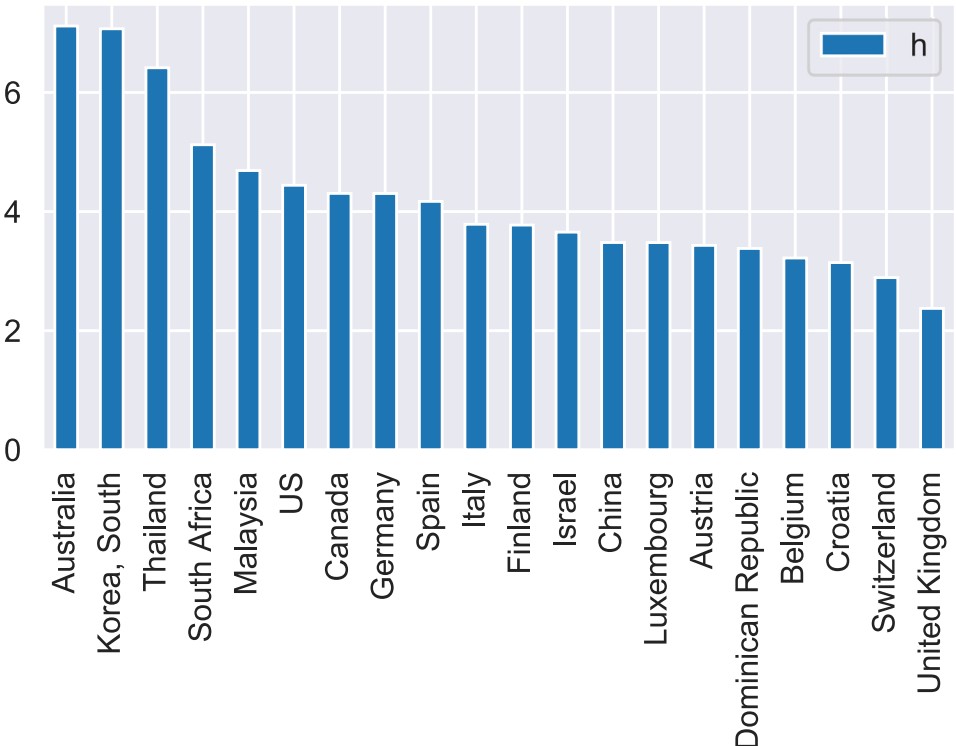

**Fig 2. Ranking of countries (top 20) with more that 1000 cases according to the calculated value of the *h* exponent.**
Data up to 4-4-2020.

Fig 3 is a geographic plot of the *h* exponent for the 55 Countries that we have studied. See for a complete list of country names and data used for the plot. Yellow color indicates high values of the *h* exponent while blue indicates an *h* exponent close to one. Countries with less than 1000 confirmed COVID-19 cases are not included in the map.

Finally, it would be helpful to identify commonalities in the set of parameters *a*, *c*, *h*. Thus, we assume that each country is epidemiologicaly characterized by the three component vector *a*, *c*, *h*. Since Eq 4 is an approximation it is anticipated that these three parameters are correlated. Thus, we use an unsupervised learning method, namely a Principle Components Analysis (PCA), a statistical procedure that uses an orthogonal transformation to convert a set of observations of possibly correlated variables into a set of values of linearly uncorrelated variables called principal components [17]. PCA is by far the most popular feature extraction and dimensionality reduction algorithm. It is also used for exploratory data analyses. e.g. for the analysis of genome data and gene expression levels in the field of bioinformatics [18]. With PCA we try to identify patterns in data based on the correlation between features. PCA aims to find the directions of maximum variance in usually highdimensional data and projects the data onto a new subspace with equal or fewer dimensions than the original one. These orthogonal axes of the new subspace i.e. the principal components can be interpreted as the directions of maximum variance with the added benefit that the new feature axes are orthogonal to each other. A classical example in unsupervised machine learning is the study of the famous Iris database, compring 4 features, by means of PCA [19, 20]. A plot of the two principle components can be used as an unsupervised classification method as different flower species are grouped together.

Motivated by this and similar examples, in Fig 4 we present a plot of the two largest principle components resulting from performing PCA on our results i.e. assuming that each country

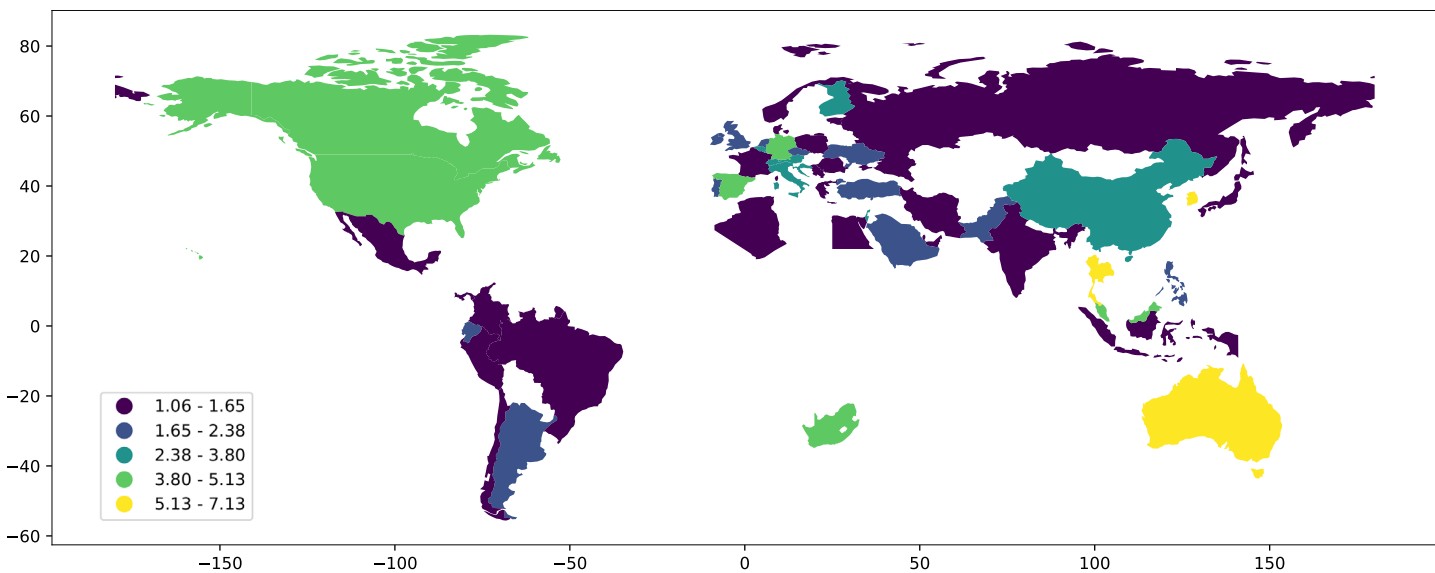

**Fig 3. Geographic plot of the *h* exponent for the 55 countries that we have studied.** See for a complete list of country names and data used for the plot.

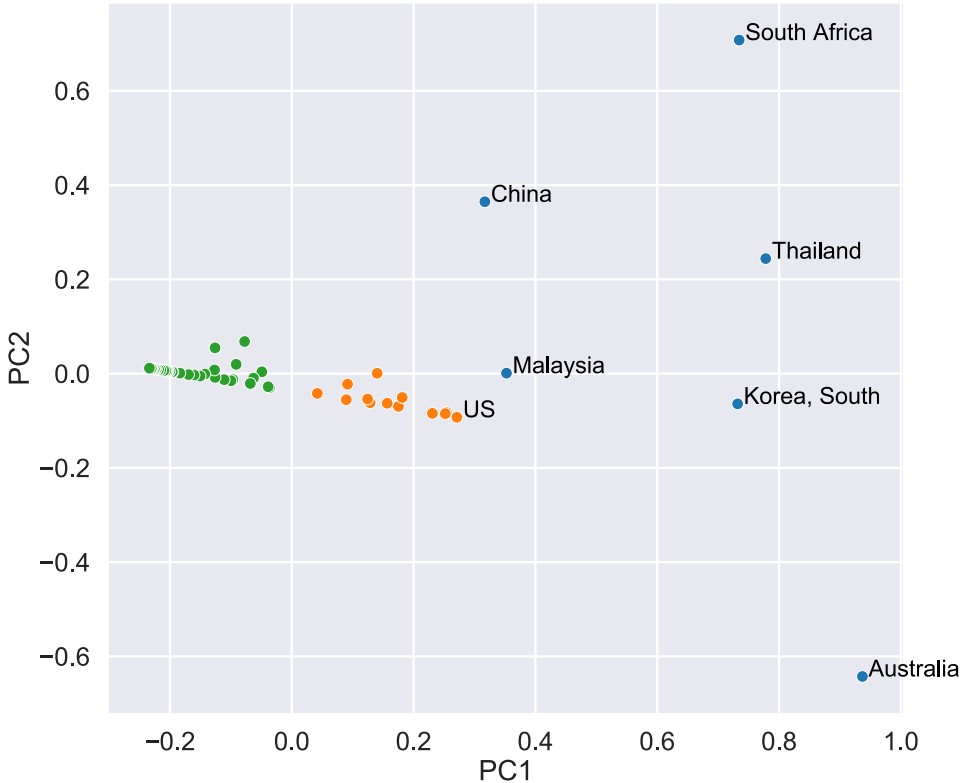

**Fig 4. Plot of the largest *PC*1 vs the second largest *PC*2 principle components for 55 countries with more than 1000 confirmed cases.** Each country is characterized by its *a*, *c*, *h* vector of the Fractal SI model. The principle components were calculated with data up to 4-4-2020. For clarity only the names of countries with $PC1 > 0.26$ are shown.

is represented by the three component vector $a$, $c$, $h$. PCA was performed using the python sci-kit-learn library [21]. Data i.e. $a$, $c$, $h$ values of the 55 countries listed in Supporting Information were preproceded by scaling the 3 parameters individually so that their respective range is between zero and one.

We observe roughly 3 clusters -colored by green, orange and blue. The majority of countries have a negative value of $PC1$ and a small value of $PC2$. There is second cluster of countries with positive $PC1$ and small $PC2$ value, including countries like US, Canada and Germany. The 3rd cluster comprises countries with high values of $PC1$ like South Korea, Malaysia and Australia which are also countries with large $h$ exponent.

## Conclusions

In this paper we introduce an extension of the classical SI model, the Fractal Kinetics SI model originating from chemical reaction kinetics. Fractal kinetic models have successfully been used in the past to describe reaction dynamics when imperfect mixing and segregation of the reactants is important and affects the dynamics of the reaction. The Fractal Kinetics SI model introduces a novel epidemiological parameter, the "fractal" exponent $h$ which is introduced in order to account for the self-organization of the societies against the pandemic through social distancing, lockdowns and flight restrictions. The major goal of this work is to propose efficient ways to model the epidemic spreading when taking under account the "heterogeneity" that arises as a response of the societies to the epidemic. The fractal kinetic framework, is a parsimonious extension of the classical models which seems to work surprisingly well.

## Supporting information

**S1 File. Covid-19 Cases_5_4.csv.** The file contains the data that we have used for the estimation of the Fractal SI model parameters. It was obtained from [14] and contains COVID-19 data up to the 5 April 2020.
(CSV)

**S2 File. API_SP.POP.TOTL_DS2_en_csv_v2_866861.csv.** The file contains population data for each country that we have studied. The 2018 population data were used in combination with the data of $\chi^2$ to calculate the fraction of COVID-19 cases per country. The file was obtained from the World Bank data repository [22].
(CSV)

**S3 File. Fit_Statistics_lmfit.docx.** The fitting of the model 2 to the data in file was performed using the python lmfit module [23]. For each country the starting values for the parameter were $a = [60, 600, 6000]$, $c = [10, 80, 800]$, $h = [2.5, 4.0]$ i.e 18 possible triplets. For each one of these 18 starting "points" we applied a basin-hopping algorithm with a Nelder-Mead local optimizer and independently a Levenberg-Marquardt minimization algorithm. (Basin-hopping is a two-phase method that combines a global stepping algorithm with local minimization at each step.) Thus, for each country we obtained 36 estimations for the values of the parameters $a$, $c$, $h$ and among them we choose the one with the minimum chi-squared $\chi^2$. Here we include the fit statistics for the countries appearing in Fig 1.
(DOCX)

**S4 File. Countries_Parameters_extended.csv.** Fitting results including estimated parameter values, chi-squared $\chi^2$, reduced chi-squared and Akaike information Criterion for the 55 Countries that we have studied and appear in Fig 4.
(CSV)

**S5 File. Countries_PCA.csv.** The PCA data for the 55 Countries appearing in Fig 4. The file contains the 3 Principle Components for each country. PCA was performed using the python scikit-learn library [21].
(CSV)

**S1 Appendix. Structural identifiability of the fractal kinetic SI model.**
(PDF)

**S1 Raw images.**
(PDF)

## Author Contributions

**Conceptualization:** Kosmas Kosmidis, Panos Macheras.

**Formal analysis:** Kosmas Kosmidis.

**Funding acquisition:** Panos Macheras.

**Investigation:** Kosmas Kosmidis.

**Methodology:** Kosmas Kosmidis, Panos Macheras.

**Software:** Kosmas Kosmidis.

**Writing – original draft:** Kosmas Kosmidis, Panos Macheras.

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
