## [Decision Letter · Decision Letter 0]

15 May 2020

PONE-D-20-10582

A Fractal kinetics SI model can explain the dynamics of COVID-19 epidemics

PLOS ONE

Dear Dr Kosmidis,

Thank you for submitting your manuscript to PLOS ONE. After careful consideration, we feel that it has merit but does not fully meet PLOS ONE’s publication criteria as it currently stands. Therefore, we invite you to submit a revised version of the manuscript that addresses the points raised during the review process.

Specifically, you are requested to comply with all the issues revised by the reviewers in a revised version of the manuscript, which may undergo a second round of review.

We would appreciate receiving your revised manuscript by Jun 29 2020 11:59PM. To enhance the reproducibility of your results, we recommend that if applicable you deposit your laboratory protocols in protocols.io, where a protocol can be assigned its own identifier (DOI) such that it can be cited independently in the future. For instructions see: http://journals.plos.org/plosone/s/submission-guidelines#loc-laboratory-protocols

We look forward to receiving your revised manuscript.

Kind regards,

Oscar Millet

Academic Editor

PLOS ONE

Journal Requirements:

2. Please consider including a Discussions and/or Conclusions section.

3. Please include in the manuscript information on the dataset used, and how this was accessed.

4. Please amend your Competing interests statement to declare any author commercial affiliations.

Reviewers' comments:

Reviewer's Responses to Questions

**Comments to the Author**

1. Is the manuscript technically sound, and do the data support the conclusions?

Reviewer #1: Partly

Reviewer #2: Partly

2. Has the statistical analysis been performed appropriately and rigorously? 

Reviewer #1: Yes

Reviewer #2: No

3. Have the authors made all data underlying the findings in their manuscript fully available?

Reviewer #1: No

Reviewer #2: No

4. Is the manuscript presented in an intelligible fashion and written in standard English?

Reviewer #1: Yes

Reviewer #2: No

5. Review Comments to the Author

Reviewer #1: In this paper, the authors adopt a quite simple fractal SI model to the cumulative data of several countries facing with COVID-19 epidemics. Basically, this is a worthwhile try, but more detailed analyses have to be carried out before acception.

(1) The fractal SI model is not fully justified. Though there is a breif mention that fractal kineitcs "describe reaction dynamics when imperfect mixing and segregation of the reactants". But more detailed connections with epidemics need to be clarfied.

(2) Statistical analysis on the model paramters and their influence on the prediction are missing.

(3) The confidence region for cumulative data is quite narrow. Daily data is more sensitive.

(4) Would the replacement of differential equation in (2) by difference equation resolve the diveregence problem, when t goes to zero?

(5) All data used in the manuscript should be made publically available, either in SI or on a webserver.

(6) PCA is an interesting study. Though the origin and consequence of three clusters appearing in Fig. 3 requires more explanations.

(7) Does h have some correlation with the basic reproduction number? What's the physical meaning of a ranking based on h in Fig. 2?

(8) A full list on the countries studied in the manuscript is needed. Otherwise, Fig. 2 and 3 become meaningless.

Reviewer #2: The idea and topic is interesting. However, there are many things missing and some caveats. The authors argue that the model fits very well to the data. However, there are many models that can fit very well to the same datasets. Somebody can argue that the best one is the one with the smallest SSR. In addition, it is not clear if the parameter values are unique. The authors need do an identifiability analysis. The authors need to explain better the justification for the parameter h.

6. PLOS authors have the option to publish the peer review history of their article (what does this mean?). If published, this will include your full peer review and any attached files.

Reviewer #1: No

Reviewer #2: No

---

## [Author Response · Author response to Decision Letter 0]

18 Jun 2020

All reviewer requests were taken care in the revised manuscript

---

## [Decision Letter · Decision Letter 1]

6 Jul 2020

PONE-D-20-10582R1

A Fractal kinetics SI model can explain the dynamics of COVID-19 epidemics

PLOS ONE

Dear Dr. Kosmidis,

Thank you for submitting your manuscript to PLOS ONE. After careful consideration, we feel that it has merit but does not fully meet PLOS ONE’s publication criteria as it currently stands. Therefore, we invite you to submit a revised version of the manuscript that addresses the points raised during the review process.

Specifically, there are still a few minor issues that need to be addressed before the manuscript becomes acceptable for publication.

We look forward to receiving your revised manuscript.

Kind regards,

Oscar Millet

Academic Editor

PLOS ONE

Reviewers' comments:

Reviewer's Responses to Questions

**Comments to the Author**

1. If the authors have adequately addressed your comments raised in a previous round of review and you feel that this manuscript is now acceptable for publication, you may indicate that here to bypass the “Comments to the Author” section, enter your conflict of interest statement in the “Confidential to Editor” section, and submit your "Accept" recommendation.

Reviewer #1: All comments have been addressed

Reviewer #2: (No Response)

2. Is the manuscript technically sound, and do the data support the conclusions?

Reviewer #1: Yes

Reviewer #2: Partly

3. Has the statistical analysis been performed appropriately and rigorously? 

Reviewer #1: Yes

Reviewer #2: No

4. Have the authors made all data underlying the findings in their manuscript fully available?

Reviewer #1: Yes

Reviewer #2: Yes

5. Is the manuscript presented in an intelligible fashion and written in standard English?

Reviewer #1: Yes

Reviewer #2: Yes

6. Review Comments to the Author

Reviewer #1: I think the authors have properly addressed all of my previous concerns in the revision. So I recommend its publication in PLoS One.

Reviewer #2: The authors improved the paper.

The PCA is not clear. Please add additional comments to understand the aims of using it.

The fitting process in appendix S3 is local. What about the global minimum?

Sturctural identifiability is only a necessary condition for “practical” identifiability; it is not sufficient. Since one of the main points of the article is the h value and the fitting process, it is important to show the practical identifiability of the 3 parameters with the available data.

7. PLOS authors have the option to publish the peer review history of their article (what does this mean?). If published, this will include your full peer review and any attached files.

Reviewer #1: No

Reviewer #2: No

---

## [Editor Report · Decision Letter 2]

27 Jul 2020

A Fractal kinetics SI model can explain the dynamics of COVID-19 epidemics

PONE-D-20-10582R2

Dear Dr. Kosmidis,

We’re pleased to inform you that your manuscript has been judged scientifically suitable for publication and will be formally accepted for publication once it meets all outstanding technical requirements.

Kind regards,

Oscar Millet

Academic Editor

PLOS ONE
---

## [Editor Report · Acceptance letter]

29 Jul 2020

PONE-D-20-10582R2 

A Fractal kinetics SI model can explain the dynamics of COVID-19 epidemics 

Dear Dr. Kosmidis:

I'm pleased to inform you that your manuscript has been deemed suitable for publication in PLOS ONE. Congratulations! Your manuscript is now with our production department. 

Kind regards, 

on behalf of

Dr. Oscar Millet 

Academic Editor

PLOS ONE